# Hardware and Software Platform Inference

Cheng Zhang [* 1]    Hanna Foerster [* 2]    Robert D. Mullins [2]    Yiren Zhao [1]    Ilia Shumailov [3]

## Abstract

It is now a common business practice to buy access to large language model (LLM) inference rather than self-host, because of significant upfront hardware infrastructure and energy costs. However, as a buyer, there is no mechanism to verify the authenticity of the advertised service including the serving hardware platform, e.g. that it is actually being served using an NVIDIA H100. Furthermore, there are reports suggesting that model providers may deliver models that differ slightly from the advertised ones, often to make them run on less expensive hardware. That way, a client pays premium for a capable model access on more expensive hardware, yet ends up being served by a (potentially less capable) cheaper model on cheaper hardware. In this paper we introduce *hardware and software platform inference (HSPI)* – a method for identifying the underlying `GPU` architecture and software stack of a (black-box) machine learning model solely based on its input-output behavior. Our method leverages the inherent differences of various `GPU` architectures and compilers to distinguish between different `GPU` types and software stacks. We evaluate HSPI against models served on different real hardware and find that in a white-box setting we can distinguish between different `GPU`s with between $83.9\%$ and $100\%$ accuracy. Even in a black-box setting we achieve results that are up to $3\times$ higher than random guess accuracy. Our code is available at https://github.com/ChengZhang-98/HSPI.

## 1. Introduction

The widespread adoption of large language models (LLMs) has transformed the technological landscape, integrating machine learning models across various sectors. However, deploying these powerful models often entails substantial upfront investments in specialized hardware infrastructure and energy, leading many businesses to opt for third-party LLM providers. This practice raises concerns about transparency and accountability, as ***buyers currently lack the means to verify the actual hardware used to serve the models they purchase***. Moreover, reports have emerged suggesting that some providers may deploy models that deviate subtly from their advertised counterparts, potentially optimized for less expensive hardware to reduce costs[1].

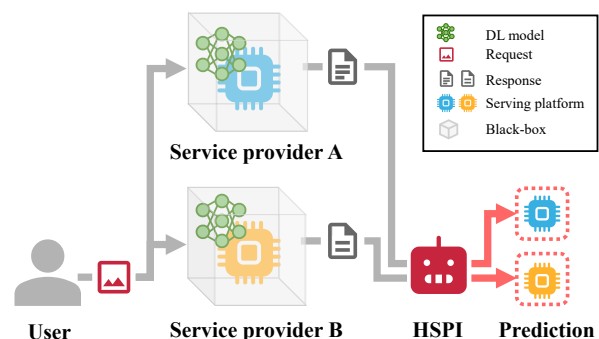

*Figure 1.* Overview of hardware and software platform inference (HSPI). HSPI aims to identify the underlying hardware and software platform of deep learning models. Engineered requests are sent to a service provider and responses are collected. With only the responses, HSPI predicts information on the hardware and software supply chains of the service provider.

A difference in hardware could not only introduce performance differences in terms of run time and model accuracy but may also indicate other potential issues. A malicious provider might employ poorer security measures, for example by running a `GPU` without a TEE present or deploying the `GPU`s at a restricted geographical location that is different from the agreed one. There might also be cases of a man-in-the-middle that is conning both the service provider

---

[*]Equal contribution    [1]Department of Computing, Imperial College London, London, United Kingdom [2]Department of Computer Science and Technology, University of Cambridge, Cambridge, United Kingdom [3]Google DeepMind, London, United Kingdom. Correspondence to: Cheng Zhang <cheng.zhang122@imperial.ac.uk>.

*Proceedings of the 42nd International Conference on Machine Learning*, Vancouver, Canada. PMLR 267, 2025. Copyright 2025 by the author(s).

---

[1]For example, here is a provider discussing strategies of reducing costs in model serving including changing models appropriately for smaller hardware.

and the client by using the service provider's service for oneself and serving a counterfeit `GPU` to a client. Moreover, a malicious provider could be after the client's prompts or data, leading to privacy concerns. ***Therefore, being able to identify a serving hardware or software platform can serve as a useful signal for a variety of reasons***.

This paper introduces **hardware and software platform inference (HSPI)** for machine learning, a novel problem formulation for identifying the underlying `GPU` architecture and potentially the software stack of a (black-box) machine learning model solely by examining its input-output behavior. HSPI works by exploiting subtle differences in how different GPUs and software environments perform calculations, which result in unique subtle patterns in the model's output. By analyzing these numerical patterns, our proposed classification framework can accurately discern the specific device employed for model inference. HSPI has significant implications for ensuring transparency and accountability. By enabling buyers to independently verify the hardware used by their providers, HSPI can help establish trust and prevent potential cost-saving measures that might compromise model performance.

To this end, we introduce two methods: HSPI with Border Inputs (HSPI-BI) and HSPI with Logits Distributions (HSPI-LD). We demonstrate the efficacy of these techniques in HSPI under both white-box and black-box setups, extending across both vision and language tasks. Overall, we make the following contributions:

- We define Hardware and Software Platform Inference (HSPI), detailing the underlying assumptions.

- We introduce two methods: HSPI with Border Inputs (HSPI-BI) and HSPI with Logit Distributions (HSPI-LD), and show their near-perfect success rates in white-box settings distinguishing between quantization levels and high accuracy rates of between $83.9\%$ and $100\%$ for distinguishing between real hardware platforms. Our empirical findings also indicate success rates in certain black-box setups that are up to three times higher than random guess accuracy. We experiment with both emulated quantization and real `GPU` hardware.

- We describe the limitations of HSPI in white- and black-box setups and provide a discussion on their potential usage across software and hardware supply chains for transparency and ML governance.

## 2. What makes HSPI possible?

We briefly explain how varying software and hardware configurations can shift a model into different *Equivalence Classes*, and how arithmetic ordering and optimizations can contribute to computational discrepancies.

**Equivalence Classes**: Different hardware and software configurations give us various computational results. When the computational results stay the same and do not deviate between settings, we talk about them being in the same equivalence class (EQC). EQCs are used to group similar computational behaviors that yield consistent results under specific settings, such as quantization levels, `GPU` architectures, and batch sizes. Schlögl et al. (2023) provide a more granular analysis, examining EQCs layer by layer across models, identifying how architectural choices impact computational stability and precision deviations.

**Factors Influencing Computational Deviations**: Schlögl et al. also examine several reasons for precision deviations, including arithmetic units using faster approximations, intermediate values being rounded to fit in registers of different sizes, transformations made during execution planning, and various tricks that optimize performance in ML toolboxes such as loop unrolling, constant folding, and arithmetic simplifications (2021; 2023). Here is a simple example of different results when decimals are rounded to integer ($\lfloor \cdot \rfloor$) before addition:

$$\lfloor \lfloor (100.4 + 0.4) \rfloor + 0.5 \rfloor = \lfloor 101 + 0.5 \rfloor = 102 \quad (1)$$
$$\lfloor 100.4 + \lfloor (0.4 + 0.5) \rfloor \rfloor = \lfloor 100.4 + 1 \rfloor = 101. \quad (2)$$

Due to finite numerical precision and stochastic pipelining, such examples can be found at any quantization level. Quantization is a technique for improving hardware efficiency at inference time. Additionally, Schlögl et al. (2023) find that the extent of deviations in computations is influenced by neural network architectures, *e.g.*, layers involving high multiplication counts tend to amplify deviations.

## 3. Related Work

**Hardware Fingerprinting in ML:** Schlögl et al. first identify unique hardware fingerprints, discovering that different hardware platforms are in different EQCs (2021; 2021). They propose boundary samples, which are inputs to a model that output results at the decision boundary between two classes. For example, an input can lie at the boundary of being classified as cat or dog, depending on the hardware. They use an adaptation of the search algorithm iterative fast-gradient-sign-method (FGSM), also known as Projected Gradient Descent (PGD), for generating adversarial samples and differentiate between 4 CPU models.

This sets the groundwork for our hardware fingerprinting methods. However, their algorithm is inefficient due to the two phase setup and the remote phase has a success rate of only $28.25\%$ on CIFAR10. Our work improves this method with HSPI-BI, and we additionally propose a new method without the need of backward propagation (HSPI-LD). Further, we mainly target `GPUs`, considering both white-box setup and black-box setup, and extensively

perform evaluations on modern ML workloads like large language models.

**Exploitation of Floating Point Inaccuracies:** Some work has also emerged exploiting floating point inaccuracies, for example, against robustness verifiers of neural networks (Jia & Rinard, 2021; Zombori et al., 2021). They point out that errors in floating point can be used against verifiers that do not consider deviations in floating point computations. Zombori et al. suggest adding small perturbations to the weights as an adhoc mitigation, they warn against other attacks using these deviations. As such, Clifford et al. for instance show that deviations in how specific operations are calculated on different hardware platforms can enable locking models to certain hardware (Clifford et al., 2024).

**Black-box LLM Model Identification:** Other works have focused on identifying the LLM model family in a black-box setting with only access to output text through semantic analysis (Iourovitski et al., 2024; Pasquini et al., 2024; McGovern et al., 2024), a variant where logit outputs are assumed (Yang & Wu, 2024) or a more white-box setting where outputs from each layer can be accessed (Zeng et al., 2023; Zhang et al., 2024). Accuracies for distinguishing between model families in the black-box setting have been 72% and 95% with only 8 interactions (Iourovitski et al., 2024; Pasquini et al., 2024).

## 4. Methodology

We start by formally describing the HSPI and listing the underlying assumptions in Section 4.1. In Section 4.2, we propose *border inputs*, which are inputs that are specifically designed to elicit divergent behavior in varying hardware and software configurations. In Section 4.3, we devise another method without the need of specifically-crafted data, by identifying deviations in floating-point distribution of model-returned logits.

### 4.1. Problem Formulation and Assumptions

The implementation of model serving in practical settings differs in accessibility. Service providers like Google, OpenAI, or Anthropic keep the underlying model architecture undisclosed, and users simply send queries and receive responses through API calls. Most services, including OpenAI APIs, are capable of providing users with the model output probabilities (logits). In other scenarios, we have cloud providers, such as Azure and AWS, for the deployment of open-source models, where both the model weights and architectures are known. We thus evaluate the following two representative scenarios:

- **White-box access to the model** – the deployed model is known and can be accessed freely. For example,

publicly available LLMs, *e.g.*, LLaMA (Dubey et al., 2024) and Gemma (Team et al., 2024), can be deployed locally for sending queries and receiving responses;

- **Black-box access only** – the deployed model can only be accessed via cloud-based interface, where it returns the output probabilities. This is similar to the current serving practices of Google, OpenAI, and Anthropic.

We also assume that it is possible to access $N$ different hardware platforms ($\mathcal{H} = \{H_0, H_1 ..., H_{N-1}\}$), where HSPI then tries to identify the hardware $H_t \in \mathcal{H}$ that the model is deployed on. To the best of our knowledge, currently, all known hardware platforms suitable for model serving can be rented. Even newer hardware devices eventually are becoming accessible on demand as seen with recent Groq (Gwennap, 2020) and Cerebras (Lie, 2022) hardware, thus making our approach feasible and realistic.

Note that slight performance and numerical variations may arise from the underlying software stack, even though the algorithm and hardware remain constant. For example, in context of machine learning, GPU kernel fusion strategies and runtime scheduling can influence the EQC of the model. We also explicitly include these variations in our $\mathcal{H}$. Consequently, HSPI can also be used to determine both the hardware platform and the software configuration, thereby identifying the combined hardware-software supply chain.

### 4.2. HSPI-BI: HSPI with Border Inputs

We reintroduce the concept of boundary samples with the name *border inputs*. As explained by Schlögl et al. these are specially crafted inputs that are at the decision boundary between two or more output classes of a model (2023). The idea of border inputs is similar to the idea of adversarial examples and we also modify PGD to create border inputs, however, formulate our loss function differently.

Specifically, consider a model $F$ which runs in two different hardware environments. Deploying models on alternative hardware environments $H$ or $H'$ can lead to subtle divergences in logit outputs $F_H$. To maximize this divergence, we modify input $X$ until the predicted labels $y$ and $y'$ differ. We define a loss function $L_{pgd}$ as the sum of cross-entropy losses between each model's logits and the other's predicted labels:

$$\mathcal{L}_{pgd} = \mathcal{L}(F_H(X), y') + \mathcal{L}(F_{H'}(X), y). \qquad (3)$$

We maximize $L_{pgd}$ by iteratively updating $X$ along the gradient direction while clamping it within a valid range of the original input, thereby encouraging the models to predict different class labels.

We can also maximize divergence by pushing $F_H(X)$ toward a target class $y_t$ while pushing $F_{H'}(X)$ away from it.

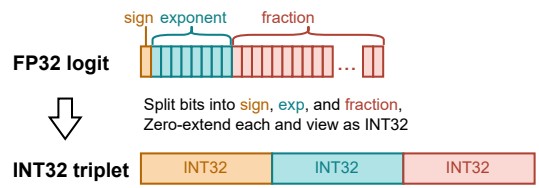

*Figure 2.* Splitting an FP32 logit into three INT32 numbers. In case that rounding noise pollutes the bit distribution in FP32 logits, before training SVMs, for each logit, we extract the sign, exponent, and fraction, zero pad each component and view each as an integer.

This is achieved through a loss function that subtracts the cross-entropy losses:

$$\mathcal{L}_{\text{1-vs-1, targeted}} = \mathcal{L}(F_H(X), y_t) - \mathcal{L}(F_{H'}(X), y_t). \quad (4)$$

When the number of possible serving platforms $N$ is small, we can extend this to compare one model against all others by summing all losses and subtracting the loss only for the one model for which we want a distinct class:

$$\mathcal{L}_{\text{1-vs-rest}} = \left( \sum_{k \in \mathcal{N} \setminus \{i\}} (\mathcal{L}(F_{H_i}(X), y_t)) \right) - \mathcal{L}(F_H(X), y_t)). \quad (5)$$

As $N$ increases, optimizing between models becomes more challenging. In such cases, we decompose the HSPI task as $\frac{N(N-1)}{2}$ binary classification problems by applying Equation (3) or Equation (4) iteratively across pairs of models to estimate the most likely hardware environment probabilistically. Note that when testing various precomputed border inputs in a black-box setting, logits become unnecessary and only the predicted class is needed.

### 4.3. HSPI-LD: HSPI with Logit Distributions

An alternative approach involves developing a classification model leveraging the information in the distribution of output logits. The logits of a set of inputs reveal characteristics of the hardware environment. The logit distribution is especially informative when inputs are diverse in the distribution of classes and closeness to class boundaries. To amplify these characteristics, we convert the logits into binary representations for small models, or more cost-efficiently, split the floating-point components (Figure 2) for large models.

We find models running on different hardware configurations produce distinct bit patterns in their logits, with certain bits used more or less frequently. Given access to all hardware configurations $\mathcal{H}$ and input samples $X_i \in \mathcal{X}$, we can create a classifier $G$ that learns to identify the environment (e.g., whether a sample was processed by $F_H$ or $F_{H'}$). Mathematically, this classifier can be defined as:

$$G(F(X_i)) = \begin{cases} 1, & \text{if } F = F_H \\ 0, & \text{if } F = F_{H'} \end{cases}. \quad (6)$$

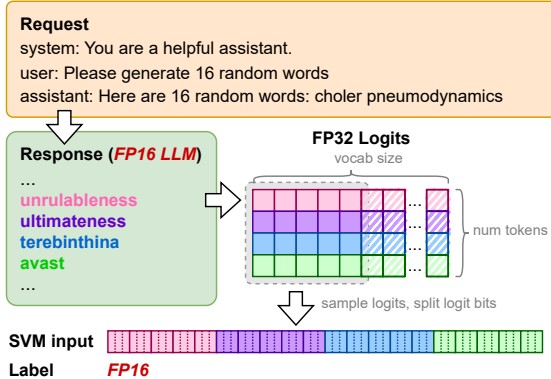

*Figure 3.* Generating HSPI-LD samples using LLMs. We guide LLMs to generate random words. The logits are flattened to form an input vector for training hardware platform classifiers.

In practice, we use an SVM as the classifier $G$ and train this classifier on a small calibration set of logits.

### 4.4. Model-Specific Adaptation

We adapt the general methods defined in Section 4.2 and Section 4.3 to fit vision models or language models. For vision models, the border inputs of HSPI-BI are randomly sampled from the model's training set, while the input images of HSPI-LD are random noise as they trigger more diverse areas of the feature space. We also carefully round border images or noise images to integers between 0 and 255 to ensure the final images are available in PNG format.

For language models, we generate random texts as the initial border queries of HSPI-BI. Since input IDs are one-hot encoded, we need to ensure that the updated border requests are still valid input IDs at the start of each PGD step. To achieve this, we first update the one-hot encoded input IDs, then use argmax to find the index of the maximum value in the updated vector for each token. For the HSPI-LD method, we guide the model to generate random words and sample top-$k$ logits of each token. We find such prompts encourage the top-$k$ logits to have closer values and avoid -Inf values. Figure 3 shows how output logits are sampled.

## 5. Evaluation

In this section, we briefly describe the experiment setup in Section 5.1, and then present the results of white-box in Section 5.2 and black-box attacks Section 5.3.

### 5.1. Experiment Setup

We consider the white-box and black-box attack setups described in Section 4.1. We build the set of hardware $\mathcal{H}$ under two distinct configurations: (1) Initially, we examine models in low-precision formats (quantization), a popular technique integrated into frameworks like HuggingFace (Wolf et al.,

2020), TensorRT (NVIDIA, 2024), and TorchAO (Torchao, 2024). Hardware devices typically feature specific arithmetic operators (*e.g.*, INT4 for A100 and FP8 for H100), and we regard differentiating different low-precision formats as a preliminary step before differentiating actual GPUs. (2) We then extend the experiments to a setup closer to real deployment scenarios, comparing actual GPUs to operate with various arithmetic configurations, different kernel implementations, and across varying device families.

**Vision Models**   We mainly use the image classification models from torchvision and fine-tune them on CI-FAR10, including ResNet18, ResNet50 (He et al., 2016), VGG16 (Simonyan & Zisserman, 2014), EfficientNet-B0 (Tan & Le, 2019), DenseNet-121 (Fu et al., 2021), MobileNet-v2 (Sandler et al., 2018), MobileNet-v3-small, and MobileNet-V3-large (Howard et al., 2019). We short-list the number formats, and GPU specs in Appendix B.1. Note that not all GPUs were used for all groups of experiments due to different server locations and the difficulty of connecting all of them for the attack generation phase.

**Language Models**   We use the open-sourced instruction-tuned LLMs from HuggingFace, including Distill-GPT2 (Sanh et al., 2019), LLaMA-3.1 (Dubey et al., 2024), QWen-2.5 (Yang et al., 2024), Phi-3.5 (Abdin et al., 2024), Gemma-2 (Team et al., 2024), and Mistral (Jiang et al., 2023). We use the official chat templates to guide the LLM to generate random words. The prompts also include a few manually added random words to increase the diversity of generation. Detailed prompts can be found in Appendix B.2. Besides standard floating-point formats, we adopt two SoTA quantization methods highly optimized for LLMs. The detailed configurations and GPU specs can also be found in Appendix B.2.

### 5.2. White-box Attacks

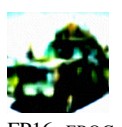 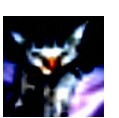 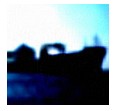 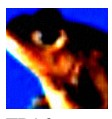

FP16: FROG  FP16: PLANE  FP16: PLANE  FP16: FROG
INT8: PLANE  MXINT8: DEER  FP8-E4: CAT  FP8-E3: CAT

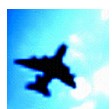 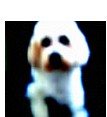 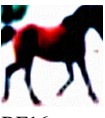 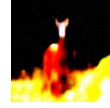

BF16: CAT  BF16: DOG  BF16: HORSE  BF16: DEER
INT8: PLANE  MXINT8: BIRD  FP8-E4: CAT  FP8-E3: CAT

*Figure 4.* Example border images of MobileNet-v3-Small generated by HSPI-BI. The predicted label changes when fed to the same model quantized to different number formats. The subcaption follows the format of model format : predicted label.

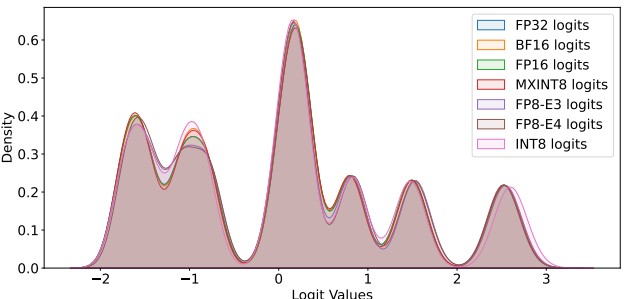

*Figure 5.* Kernel density estimate of logit distributions of different quantization classes on the classification of the same 5000 images for CIFAR10 with ResNet18, i.e., 50000 logits.

**Vision Models**   We find HSPI-BI always distinguishes between different number formats/quantization methods on the same GPU. Examples of border images can be found in Figure 4. Figure 5 illustrates the kernel density estimation of various quantization methods where the shapes imply the obvious differences. When extending to real GPUs, we find the HSPI-BI generally works except for GPUs with very similar specs. For example, border images can be created between RTX8000 and H100 when running on identical CUDA version and PyTorch version, but not between RTX8000 and RTX2080Ti (see Table 8 in the appendix). This is because both RTX8000 and RTX2080Ti have the same Compute Capability of 7.5 and a very similar number of Tensor Cores around 550.

HSPI-LD can distinguish between all quantization levels for the models ResNet18, ResNet50, VGG16, EfficientNet, DenseNet, and MobileNet-V2 when using an SVM with a set of 10 images' logits per sample. When comparing across GPUs, HSPI-LD are able to distinguish the A100 well from the RTX8000 and RTX2080Ti with an accuracy of 100% (Figure 9 in the appendix shows the clear difference of logit bit distribution). Again, HSPI-LD fails to distinguish between the RTX8000 and RTX2080Ti, as calculations are in the same EQC. We discuss the limitations and robustness of HSPI in Appendix E and Appendix F.

**Language Models**   We find that HSPI-BI is not suitable for LLMs. With enough reruns and PGD iterations, HSPI-BI can generate border requests that lead to different responses across quantization methods. However, we find that the PGD process is unstable due to the discontinuity in its projection. Moreover, limited GPU memory constrains our ability to scale experiments to larger LLMs and bigger request batch sizes. This is because the forward pass of the target model is part of the computation graph for gradient descent, requiring caching the intermediate activations of LLMs in memory for backward propagation.

On the contrary, HSPI-LD achieves remarkable success in both quantization experiments and real GPU experiments.

*Table 1.* Initial white-box experiments on actual GPUs. With a fixed CUDA version, we perform HSPI-LD on LLaMA-3.1-8B. The trained SVM distinguishes GPUs, arithmetic modes, and even kernel implementations in Group 1, 2, and 3 respectively.

| Group | GPUs | Ariths. | Kernels | Acc. |
|-------|------|---------|---------|------|
| 1 | A100, A6000 | FP16 | Plain | 1.00 |
| 2 | A100 | FP16, BF16 | Plain | 1.00 |
| 3 | A100 | FP16 | Plain, FlashAttn2 | 1.00 |

*Table 2.* White-box experiments on actual GPUs. We perform white-box HSPI-LD on a setup mixing GPUs, arithmetic modes, and kernel implementations. We treat this as a classification task with 18 unique labels. By-class accuracy and F1 score are in the last two columns. HSPI-LD achieves an overall accuracy of 83.9% using only 256 requests (random guess accuracy = 5.6%).

| GPUs | Ariths. | Kernels | Class Idx | Acc. | F1. |
|------|---------|---------|-----------|------|-----|
| A100 | FP16 | Plain | 1 | 0.742 | 0.745 |
| | | SDPA | 2 | 0.645 | 0.683 |
| | | FlashAttn2 | 3 | 0.695 | 0.698 |
| | BF16 | Plain | 4 | 1 | 1 |
| | | SDPA | 5 | 1 | 1 |
| | | FlashAttn2 | 6 | 1 | 1 |
| A6000 | FP16 | Plain | 7 | 0.680 | 0.705 |
| | | SDPA | 8 | 0.656 | 0.656 |
| | | FlashAttn2 | 9 | 0.688 | 0.674 |
| | BF16 | Plain | 10 | 1 | 1 |
| | | SDPA | 11 | 1 | 1 |
| | | FlashAttn2 | 12 | 1 | 1 |
| L40S | FP16 | Plain | 13 | 0.688 | 0.689 |
| | | SDPA | 14 | 0.644 | 0.622 |
| | | FlashAttn2 | 15 | 0.664 | 0.636 |
| | BF16 | Plain | 16 | 1 | 1 |
| | | SDPA | 17 | 1 | 1 |
| | | FlashAttn2 | 18 | 1 | 1 |
| **Average** | | | | 0.839 | 0.839 |

For quantized LLaMA-3.1-8B, Phi-3.5-mini, and QWen-2.5-3B deployed in FP16, BF16, INT8-FD, and HQQ-4bit, an accuracy of over 99.5% for each model is achieved (See Table 9 in the appendix). For real `GPU` experiments, we firstly perform three experiments in Table 1 to verify that HSPI-LD can distinguish between different `GPUs`, arithmetic modes, and kernel implementations. We then conduct a comprehensive experiment in Table 2, mixing all these factors. We treat this experiment as a classification task with 18 unique labels. Remarkably, HSPI-LD achieves an overall accuracy of 83.9% (random guess accuracy = 5.6%). We visualize the evident difference in logit bit distribution between RTXA6000 and A100 in Figure 6.

Note that HSPI-LD is suitable for language models given the size of LLMs. HSPI-LD only runs inference on the model,

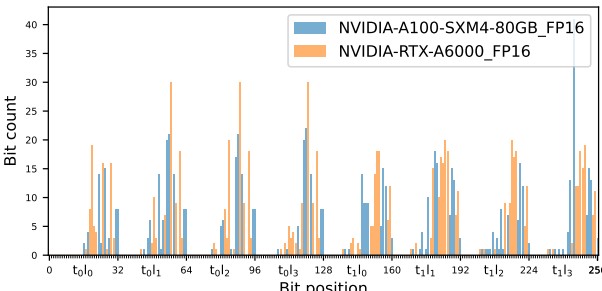

*Figure 6.* The difference of bit distribution between RTXA6000 and A100 (white-box HSPI-LD). We send the same 256 queries to QWen-2.5-3B deployed on RTXA6000 and A100 respectively and compare the bit distribution of FP32 log probabilities. Tokens and logits are sampled in the plot but the difference is still obvious. $t_i l_j$ denotes the log probability of $i$-th token's $j$-th logit.

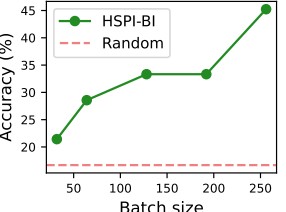 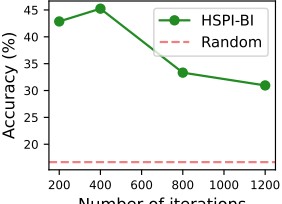

(a) Transferability vs batch size    (b) Transferability vs num iters

*Figure 7.* Transferability of border images trained on MobileNet-v3-Small. (a) The transferability is improved with larger batch sizes. (b) Given a batch size (256 in the plot), overfitting happens if the border images are trained with too many iterations.

consuming much less memory. Besides, the collection of responses (logits) for each `GPU` is independent, without the need of across-node communication.

### 5.3. Black-box Attacks

**Vision Models**    For HSPI-BI, we estimate the transferability of a set of border images by training the images on one model and testing them on other unseen models. Table 3 contains three experiments following this setup. The border images trained on MobileNet-v2 achieve an average F1-score of 0.345 across 7 unseen models (random guess F1-score = 0.144). We also have the following observations:

(1) If border images are trained in batches, a larger batch size will improve the transferability across models. Figure 7a sweeps the batch size of border images trained on MobileNet-v3-Small and plots the resultant transferability. With the increasing batch sizes, the transferability is enhanced. However, we cannot further increase the batch size after 256 due to limited `GPU` memory.

(2) Overfitting happens if the border images are trained with

*Table 3.* Transferability of HSPI-BI and HSPI-LD on quantized vision models. We consider a set of models including VGG-16, ResNet-18, ResNet-50, MobileNet-V2, EfficientNet-B0, and DenseNet-121. Each row trains the SVM classifier on one model and evaluate the transferability in terms of F1-Score on the rest models. Random guess F1-score is 0.144. For HSPI-BI, all the experiments were run on RTXA6000 with 400 iterations, and HSPI-LD experiments were run on Nvidia Quadro RTX 8000.

| Method | Training Model | Test Model | FP32 | BF16 | FP16 | MXINT8 | FP8-E3 | FP8-E4 | INT8 | Avg. F1. |
|---|---|---|---|---|---|---|---|---|---|---|
| HSPI-BI | VGG16 | Other models | 0 | 0 | 0.167 | 0.234 | 0.159 | 0.253 | 0.218 | 0.147 |
| | ResNet18 | Other models | 0 | 0 | 0.25 | 0.293 | 0.286 | 0.167 | 0.286 | 0.206 |
| | MobileNet-v2 | Other models | 0.235 | 0.345 | 0.218 | 0.167 | 0.286 | 0.444 | 0.444 | 0.345 |
| HSPI-LD | VGG16 | Other models | 0.394 | 1 | 1 | 0 | 0.95 | 0.65 | 0.2 | 0.599 |
| | ResNet18 | Other models | 0.332 | 1 | 0.986 | 0.318 | 0.972 | 0.682 | 0.446 | 0.677 |
| | ResNet50 | Other models | 1 | 1 | 1 | 0.056 | 0.602 | 0.634 | 0.642 | 0.562 |
| | MobileNet-v2 | Other models | 0.026 | 0.8 | 0.8 | 0.342 | 0.768 | 0.69 | 0.498 | 0.561 |
| | EfficientNet | Other models | 0 | 1 | 1 | 0 | 0.2 | 0.612 | 0.592 | 0.486 |
| | DenseNet-121 | Other models | 0.102 | 1 | 0.996 | 0.34 | 0.926 | 0.514 | 0.638 | 0.645 |

too many iterations. Figure 7b presents the transferability of the same group of border images trained with various number of iterations. The decreasing transferability at 800 and 1200 iterations indicates that the border images overfit the training model.

(3) The border images trained on a more complex model tend to have better transferability. For example, in Table 3, the border images trained on MobileNet-v2 has a higher F1-Score than those trained on ResNet18 and VGG16. However, again, more complex models require more `GPU` memory during training, which leads to limited scalability of HSPI-BI.

For HSPI-LD, we compare how well an SVM trained on one model performs on all other models in Table 3. We observe that the transferability in detecting BF16 and FP16 is particularly high, while for other quantization levels such as FP32 and MXINT8 the transferability is very low. Furthermore, ResNet18, followed by DenseNet perform best as models with high transferability to other models.

**Language Models**  We only perform HSPI-LD in the black-box setup for language models as the HSPI-BI method is unstable. We train an SVM on a set of training models $\mathcal{F}_s$ and test it on another set of unseen models $\mathcal{F}_t$. The results of quantized models are shown in Table 4. The transferability measured with average accuracy is between 22.7% and 60.3% (random guess accuracy is 25%). We find that overfitting also occurs if the SVM is trained with too many iterations, and the transferability heavily depends on the choice of models in the training set.

Here we summarize the promising approaches and challenges in Table 5 based on our experiments and observations. Though we achieve three to five times higher accuracy than random guess in the black-box setup, more effective solutions still need to be explored.

*Table 4.* Transferability of black-box HPI-LD on quantized LLMs. We split a set of LLMs, $\mathcal{F} = \{$`Gemma-2-2B`, `Gemma-2-9B`, `LLaMA-3.1-8B`, `Phi-3.5-mini`, `Mistral-7B-v0.3`, `QWen-2.5-1.5B`, `QWen-2.5-3B`, `QWen-2.5-7B`$\}$, into a supportive (training) set $\mathcal{F}_s$ and a test set $\mathcal{F}_t := \mathcal{F} - \mathcal{F}_s$. The SVM classifier is trained using responses from $\mathcal{F}_s$, and tested on $\mathcal{F}_t$. Random guess accuracy is 25%. We use $f_i$ ($i = 0 \ldots, 7$) to denote $i$-th model in $\mathcal{F}$.

| $\mathcal{F}_t$ | FP16 | BF16 | INT8-FD | HQQ-4bit | Avg. Acc. | Avg. F1. |
|---|---|---|---|---|---|---|
| $f_3$ | 0.0 | 0.0 | 0.865 | 0.041 | 0.227 | 0.108 |
| $f_5$ | 1.0 | 0.074 | 0.293 | 0.783 | 0.538 | 0.487 |
| $f_7$ | 0.369 | 0.969 | 0.980 | 0.093 | 0.603 | 0.545 |
| $f_5, f_7$ | 0.853 | 0.274 | 0.450 | 0.797 | 0.594 | 0.582 |

*Table 5.* Promising approaches (✓) and challenges (!) of HSPI.

| Setup | Vision Models | | Language Models | |
|---|---|---|---|---|
| | White-box | Black-box | White-box | Black-box |
| HSPI-BI | ✓ | ! | ! | ! |
| HSPI-LD | ✓ | ✓ | ✓ | ! |

## 6. Discussion

### 6.1. Tests on LLM Serving Frameworks

LLM serving frameworks like vLLM (Kwon et al., 2023) and SGLang (Zheng et al., 2024) facilitate the deployment of LLMs in production environments. In Section 5, we run the experiments with a naive setup. To verify HSPI-LD in production environments, we launch SGLang on cloud GPUs, send queries and collect responses through OpenAI API, and perform HSPI-LD. Though SGLang enables a series of optimizations, such as RadixAttention and cache-aware scheduling policy, HSPI-LD successfully recognizes different GPUs combined with various data types, partitioning strategies, and kernel implementations. Table 6 presents the results where the over 99% accuracy highlights the ef-

*Table 6.* White-box experiments of SGLang using cloud GPU service. We perform white-box HSPI-LD on a setup close to production environments where LLM serving framework, SGL, is deployed on a multi-GPU cloud server. We send queries and collect responses through OpenAI API.

| GPUs | Ariths. | Kernels | Sharding | ClassIdx | Acc. | F1. |
|---|---|---|---|---|---|---|
| L40 | FP16 | FlashInfer | DP2, TP2 | 1 | 0.961 | 0.972 |
| | | | DP4, TP1 | 2 | 1 | 0.992 |
| | | Triton | DP2, TP2 | 3 | 0.992 | 0.9988 |
| | | | DP4, TP1 | 4 | 1 | 0.992 |
| | BF16 | FlashInfer | DP2, TP2 | 5 | 1 | 1 |
| | | | DP4, TP1 | 6 | 1 | 1 |
| | | Triton | DP2, TP2 | 7 | 1 | 1 |
| | | | DP4, TP1 | 8 | 1 | 1 |
| A40 | FP16 | FlashInfer | DP2, TP2 | 9 | 1 | 0.973 |
| | | | DP4, TP1 | 10 | 0.992 | 0.988 |
| | | Triton | DP2, TP2 | 11 | 0.961 | 0.953 |
| | | | DP4, TP1 | 12 | 0.953 | 0.976 |
| | BF16 | FlashInfer | DP2, TP2 | 13 | 1 | 1 |
| | | | DP4, TP1 | 14 | 1 | 1 |
| | | Triton | DP2, TP2 | 15 | 1 | 1 |
| | | | DP4, TP1 | 16 | 1 | 1 |
| Average | | | | | 0.991 | 0.991 |

fectiveness in a production environment. In Appendix D, we include two more result tables on large models served on devices & software stack from various vendors like AMD, NVIDIA, and Amazon, highlighting that HSPI-LD is applicable to a more complex setup.

### 6.2. Scalability and Cost Implications of HSPI

One may question the potentially high cost of HSPI due to the large number of possible combinations of hardware and software. As shown in previous experiments, the number of classes is the product of possible hardware platforms and software dependencies, *e.g.*, $|\{L40, A40\} \times \{FP16, BF16\} \times \{FlashInfer, Triton\} \times \{DP2TP2, DP4TP1\}| = 16$ classes in Table 6. If we decompose this classification task into binary classification problems, $\frac{N(N-1)}{2} = 120$ classifiers need to be trained. However, we emphasize that one can reduce the number of classes by "merging labels". By "merging labels" we mean merging some unimportant labels (don't cares) into a new label. Take Table 6 as an example, if we label the eight classes $\{L40\} \times \{FP16, BF16\} \times \{FlashInfer, Triton\} \times \{DP2TP2, DP4TP1\}$ into a new label "L40" and the rest into "A40", we can train a two-class classifier with a high F1 score of 98.5% (See Table 12 in the appendix). Another factor determining the cost of HSPI is the expense of LLM API access for collecting training samples. As detailed in Appendix B, we

need around 60k tokens for each class, which is acceptable considering LLM API usually costs less than \$0.05 per 1k tokens (Cortenix & Bruner, 2024).

### 6.3. Limitations and Robustness

HSPI faces several practical limitations. Some configurations remain indistinguishable due to leaving computations in the same EQC - for instance, different CUDA versions on identical hardware, or similar GPUs (RTX8000 and RTX2080Ti) with matching compute capabilities. Moreover, batch size variations significantly affect logit outputs, requiring separate analysis for each common batch size. We discuss these limitations and possible strategies against the use of HSPI such as random bit flips in logits or noise injection in input data in Appendices E and F.

### 6.4. Software and Hardware Supply Chains

Software supply chain variations across runtimes, compilers, and high-level frameworks, such as PyTorch or Tensorflow, can affect result reproducibility even with identical models and hardware. This highlights the need for careful version control and standardized compilation practices to mitigate potential discrepancies on the software side. Meanwhile, hardware options have proliferated through cloud providers, ranging from various GPU vendors to specialized AI accelerators (Google TPUs, AWS Inferentia, Groq and Cerebras). The rise of model API marketplaces like OpenRouter[2] has introduced additional complexity, as these services often obscure their underlying hardware configurations. This lack of transparency in hardware platforms makes it difficult to verify security compliance and performance consistency. HSPI methods offer a potential solution for validating these hardware supply chains.

### 6.5. ML Governance

HSPI could significantly advance ML governance by enabling precise identification of hardware and software components in ML supply chains. This capability would strengthen three key areas: (1) traceability and accountability through detailed configuration tracking, (2) establishment of industry-wide standards for hardware and software documentation, and (3) improved reliability in model sharing and deployment. We will elaborate on these governance and broader impacts in our impact statement.

## 7. Conclusion

In this paper, we introduce Hardware and Software Platform Inference (HSPI) – a method for identifying the underlying hardware and software stack solely based on the

---

[2]https://openrouter.ai/

input-output behavior of machine learning models. Our approach leverages inherent quantization limitations and the arithmetic operation orders of different `GPU` architectures to distinguish between various `GPU` types and software stacks. Our findings demonstrate the feasibility of inferring this information from black-box models, underscoring its implications for ensuring transparency and accountability in the LLM market. We will open-source our logit dataset and codes once the paper is accepted.

## Impact Statement

HSPI's potential applications extend beyond hardware identification to several aspects of ML system governance and deployment. Building on our earlier discussion of traceability, standardization, and reliability benefits, this section examines the broader technical and practical implications of HSPI. We consider its effects on security auditing practices, deployment verification protocols, and potential impacts on hardware provisioning strategies.

In the perfect HSPI scenario, our methods can help identify the exact components in a model's software and hardware supply chains as mentioned in Section 6.4. This would allow users to produce and validate the software and hardware bill of materials, as suggested by recent research (Arora et al., 2022). By doing so, HSPI could become a key tool in enabling ML governance – a framework of standards and principles to ensure that an ML system operates responsibly upon deployment. The proposed HSPI problem formulation, including Schlögl et al.'s insights on EQCs, will help in shaping standards in different stages of an ML model's life cycle from testing to deployment (Chandrasekaran et al., 2021).

**Traceability and Accountability** In complex ML workflows, traceability of the exact software and hardware configurations used in model training and inference is essential for building trust and accountability. This would help in identifying the origin of specific model behavior differences, making reproduction of behavior easier. This traceability is not only crucial for debugging but also for audits, security, and mitigating risks from potential tampering or errors.

**Establishing Industry Standards** HSPI can drive the adoption of industry-wide standards for documenting and verifying hardware and software configurations in ML development. Standardized reporting of hardware architectures, library versions, and compilers would enable researchers and practitioners to replicate results reliably. This would also help in establishing ML quality control norms.

**Model Sharing and Deployment Reliability** Replicability through HSPI and newly established industry standards

can further model sharing practices. Detailed information about the training and inference environment can reduce deployment issues related to hardware and software mismatches and can help users understand potential limitations or dependencies on specific hardware or software components. This will ultimately create an environment fostering confidence in using models developed by other people.

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

## A. Additional Related Work

Side-channel have been exploited to infer information about the model and hardware For example, Hua et al. (2018) extracts information about CNN models, e.g, whether the model is AlexNet or SqueezeNet, from an FPGA-based CNN accelerator, by analyzing memory access patterns. Recent work Gongye et al. (2024) predicts information about the encrypted IPs on an FPGA-based DNN accelerator. They also recover characteristics about model architectures and extract model params. Our work aims to predict information about the SW-HW stack used for serving large deep learning models, including compiler, data types, GPU arch, parallelism strategy, etc.

Our work is different from these two papers in terms of goals, threat models (methods), and generalizability. Hua et al. (2018) and Gongye et al. (2024) assume side channel information about the HW like off-chip memory access patterns, electromagnetic traces, schematics is accessible, while we assume the SW-HW platform serves DNNs in the cloud and we only have access to the request and responses via the service provider's API. These two works test their methods on a specific FPGA accelerator, while We tried our best to test the generalizability, across model families, data types, GPU archs, parallelism strategies, etc.

## B. Detailed Experiment Setup

We elaborate the detailed experiment setup in this sections. For the version information of software, please refer to our source codes.

### B.1. Vision Model

**Number formats and GPU Specs** We perform the experiments on standard floating-point formats (FP32, FP16, BF16) and low-precision formats first, then extend to actual GPUs. The low-precision formats include MX-INT8 (Darvish Rouhani et al., 2020), FP8-E3M4 (noted as FP8-E3) (Shen et al., 2024), FP8-E4M3 (noted as FP8-E4), and dynamic INT8 (noted as INT8) (Kim et al., 2021). For actual GPUs, we include NVIDIA H100, A100, GeForce RTX2080 Ti, and Quadro RTX8000. Not all GPUs were used for all experiments due to different server locations and the difficulty of connecting all of them for the attack generation phase.

**Border Images** For HSPI-BI method, we randomly sample images from CIFAR10 as initial border images, and apply the PGD method described in Section 4.2 to update the border images. The projection step refers to the operation of scaling and rounding pixel values to integers between 0 and 255. For HSPI-LD method, we use images of size 224x224 that we create through randomly sampling floating point values between 0 and 1 and then converting them into the valid pixel range of integers between 0 and 255. These random images perform better than images from datasets such as CIFAR10 as they trigger more diverse areas of the feature space and hence produce a set of more distinct logits. To distinguish between logits with an SVM, we always use a set of 10 or 25 images' logits and since in our access model we test with exactly the same images, we report training accuracy for all results.

**Fine-tuning Hyper-parameters of Vision Models** The checkpoints of vision models in Section 5 are downloaded from torchvision [3]. Before HSPI experiments, we fine-tune these models on CIFAR10 to ensure they achieve reasonable accuracy on CIFAR10 test set. Specifically, we use

---

[3]https://pytorch.org/vision/stable/models.html#classification

SGD optimizer and linear learning rate scheduler with initial learning rate = 1e-3. The fine-tuning batch size is 128 and we fine tune all the models for 3 epochs.

**Accuracy of Quantized Models**    We show in Table 7 that all low-precision formats have negligible accuracy loss compared to the fine-tuned model. This serves as a sanity check before the HSPI experiments in case the quantization breaks the models.

*Table 7.* Accuracy of quantized vision models on CIFAR10.

| Model | FP32 | BF16 | FP16 | MXINT8 | FP8-E3 | FP8-E4 | INT8 |
|---|---|---|---|---|---|---|---|
| VGG16 | 0.882 | 0.882 | 0.881 | 0.879 | 0.876 | 0.875 | 0.875 |
| ResNet18 | 0.937 | 0.937 | 0.937 | 0.936 | 0.931 | 0.929 | 0.934 |
| ResNet50 | 0.965 | 0.965 | 0.965 | 0.963 | 0.962 | 0.959 | 0.951 |
| MobileNet-v2 | 0.936 | 0.936 | 0.936 | 0.929 | 0.928 | 0.926 | 0.928 |
| MobileNet-v3-small | 0.914 | 0.914 | 0.914 | 0.909 | 0.907 | 0.903 | 0.907 |
| MobileNet-v3-large | 0.946 | 0.946 | 0.946 | 0.943 | 0.937 | 0.934 | 0.945 |
| EfficientNet-B0 | 0.957 | 0.954 | 0.956 | 0.954 | 0.945 | 0.953 | 0.954 |
| DenseNet-121 | 0.962 | 0.961 | 0.962 | 0.954 | 0.954 | 0.949 | 0.952 |

### B.2. Language Models

**Quantization methods and GPUs**    Similar to vision models, we first run experiments of differentiating low-precision formats, then extend to actual GPUs. Since LLM quantization is more challenging than vision models, besides FP16 and BF16, we adopt two quantization methods that have been proven effective and integrated into HuggingFace: fine-grained dynamic INT8 quantization (Torchao, 2024), noted as INT8-FD, and half-quadratic quantization (Badri & Shaji, 2023), noted as HQQ-4bit. For actual GPUs, we consider NVIDIA A100, L40S, RTX A6000, and GeForce RTX3090.

**Prompts and logit datasets**    For HSPI-LD, we guide language models to generate random tokens with the adapted official chat template of each model. Figure 8 shows an example of the template of QWen-2.5, where {num_random_words} and {random_words} are parameters for generating queries. {random_words} is $n$ leading random words sampled from the "popular" corpora of Natural Language Toolkit[4], thus leaving ({num_random_words} - $n$) tokens for the LLM to generate. Throughout our experiments, we find {num_random_words} = 16 and $n = 3$ is enough for the white-box setup, while for the black-box setup, we adapt {num_random_words} = 16 and $n = 8$.

We generate 256 queries for each class, and sample logits in Tables 1, 4 and 9 because of varying vocabulary sizes of LLMs. Specifically, we sample the top-256 logits of each generated token, and we flatten the logits of 4 responses for a training sample of SVM classifier.

---

[4]Natural Language Toolkit: https://www.nltk.org/index.html

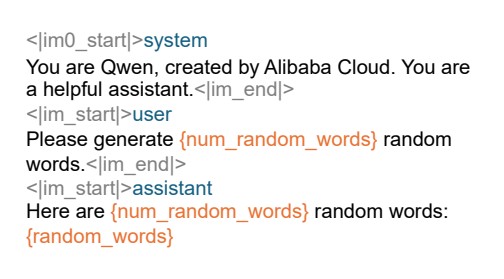

```
<|im0_start|>system
You are Qwen, created by Alibaba Cloud. You are
a helpful assistant.<|im_end|>
<|im_start|>user
Please generate {num_random_words} random
words.<|im_end|>
<|im_start|>assistant
Here are {num_random_words} random words:
{random_words}
```

*Figure 8.* The prompt template of QWen-2.5 used by HSPI-LD in Section 4.3.

## C. Additional Vision Model Results

Table 8 shows that HSPI-BI can differentiate RTX8000 and A100 but fails to differentiate RTX8000 and RTX2080Ti because they two GPUs have very similar specs. Figure 9 shows the clear difference in logit bit distribution between RTX8000 and NVIDIA A100, which explains the success of HSPI-LD.

*Table 8.* Table showing success in creating border images for different GPUs in white-box with an inference batch size of 1.

| GPUs | FP32 | BF16 | FP16 | MXINT8 | FP8-E3 | FP8-E4 | INT8 |
|---|---|---|---|---|---|---|---|
| RTX8000 vs A100 | ✓ | ✓ | ✓ | ✓ | ✗ | ✗ | ✓ |
| RTX8000 vs 2080Ti | ✗ | ✗ | ✗ | ✗ | ✗ | ✗ | ✗ |

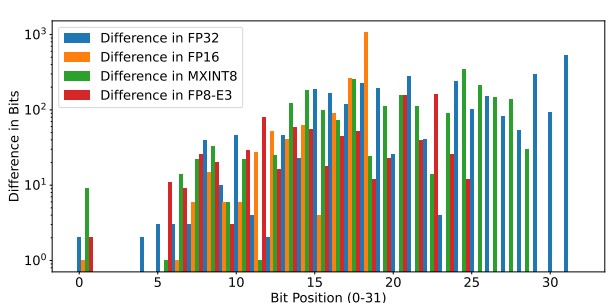

*Figure 9.* A histogram showing the difference in logit bit distribution for the classification of the same 5000 images for CIFAR10 with ResNet18, i.e., 50000 logits, between Nvidia Quadro RTX 8000 and NVIDIA A100.

## D. Additional LLM Results

### D.1. White-box HSPI-LD of quantized LLMs

Table 9 shows the results of HSPI-LD on quantization methods for LLMs with an average accuracy of 0.995.

*Table 9.* By-class accuracy of white-box HSPI-LD on quantization levels for LLMs. HPI-LD is able to differentiate LLMs deployed in FP16, BF16, INT8-FD, and HQQ-4bit. The SVM classifier is trained using responses of 256 random word requests.

| Training Model | FP16 | BF16 | INT8-FD | HQQ-4bit | Avg. Acc. |
|---|---|---|---|---|---|
| LLaMA-3.1-8B | 0.996 | 1 | 1 | 1 | 0.999 |
| QWen-2.5-3B | 1 | 1 | 1 | 1 | 1 |
| Phi-3.5-mini | 0.996 | 0.996 | 1 | 0.988 | 0.995 |

## D.2. HSPI-LD on Various GPU Systems

In Table 10, we distributed a large LLM, Llama-3.1-70B, across a larger multiple high-end GPU systems using SGLang. HSPI-LD still achieves high accuracy. We also test HSPI-LD on devices from different vendors, including NVIDIA, AMD, and Amazon in Table 11. Different devices are distinguished by their hardware architecture, software stack, and data types.

| Vendor | Hardware (arch) | N devices | DP,TP | DType | Acc | F1. |
|---|---|---|---|---|---|---|
| AMD | MI300X (CNDA3) | 2 | DP1TP2 | BF16 | 1.000 | 1.000 |
| | | | | FP16 | 0.961 | 0.957 |
| NVIDIA | H100 (Hooper) | 4 | DP2TP2 | BF16 | 1.000 | 1.000 |
| | | | | FP16 | 0.930 | 0.941 |
| | L40 (Ada) | 8 | DP1TP8 | BF16 | 1.000 | 1.000 |
| | | | | FP16 | 0.969 | 0.958 |
| | A40 (Ampere) | 8 | DP1TP8 | BF16 | 1.000 | 1.000 |
| | | | | FP16 | 0.922 | 0.925 |
| | | | | Avg | 0.973 | 0.973 |

*Table 10.* Whitebox HSPI-LD on various real GPUs.

| Vendor | Hardware | Software | DType | Accuracy | F1. |
|---|---|---|---|---|---|
| Amazon | Inferentia | NKI | BF16 | 0.968 | 0.924 |
| | | | FP16 | 0.988 | 0.908 |
| AMD | MI300X | ROCm | BF16 | 1.000 | 1.000 |
| | | | FP16 | 0.906 | 0.913 |
| NVIDIA | H100 | CUDA | BF16 | 1.000 | 1.000 |
| | | | FP16 | 0.922 | 0.915 |
| | | | Avg | 0.964 | 0.943 |

*Table 11.* White-box HSPI-LD on devices from various vendors.

## D.3. Label merging

With the same samples in Table 6, we merge the samples of the same GPU but different software dependencies into a new label and train a new classifier. Surprisingly, the classifier still achieves a high accuracy, implying that label merging can effectively reduce the number of classes and improve scalability of HSPI-LD.

*Table 12.* White-box experiments of Table 6 with merged labels. We label the samples with the same GPU but various software as the same class and train a new classifer. The classifier still achieves high accuracy and F1 score, implying label merging is an effective way to reduce the number of labels and to improve the scalability of HSPI-LD.

| GPUs | Accuracy | F1 Score |
|---|---|---|
| A40x{FP16, BF16}x{FlashInfer, Triton}x{DP2TP2, DP4TP2} | 0.990 | 0.985 |
| L40x{FP16, BF16}x{FlashInfer, Triton}x{DP2TP2, DP4TP2} | 0.979 | 0.985 |
| Average | 0.985 | 0.985 |

# E. Limitations and Black-box Failure Cases

We notice that not all software configurations have an effect on the final model performance, leaving computations in the same EQC. For example, we tested the HSPI-BI method on different CUDA versions, while keeping the hardware and the rest of the software stack fixed. A ResNet18 model trained on the CIFAR10 and run on an RTX8000 GPU, resulted in no measurable differences, thereby HSPI-BI completely failed to distinguish between the CUDA versions. Similarly, not all hardware is distinguishable. For example, HSPI-BI failed to differentiate the RTX8000 and the RTX2080Ti (both have NVIDIA Compute Capability = 7.5 and similar number of Tensor Cores around 550).

Furthermore, we see big differences in calculated logits between different inference time batch sizes. We find that the same batch size needs to be used to distinguish between hardware. This makes our methods more complicated. For example for HSPI-BI, some border images work across a few batch sizes, but do not work for all. This means that we would need to compute border inputs for all most popular batch sizes between a set of hardware platforms or at least enough to make a probable guess on it. Similarly, we would need to collect different sets of logits for various batch sizes for HSPI-LD, considering that Anthropic recently released new APIs for batched request.

This situation prompts inquiry into the nuances of software-hardware supply chains. For instance, the extent of floating-point deviations in logits varies significantly depending on the targeted supply chain. Identifying different quantization levels may be straightforward, but discerning subtle distinctions, such as those between FlashAttention V2 and V3, can present a challenge. Minor differences or distinctions in the supply chain may pose challenges to both HSPI-BI and HSPI-LD techniques. Nonetheless, there is another argument that an empirically-guided design of the loss function in HSPI-BI, or enhanced feature engineering for HSPI-LD could improve the distinction of these subtle variations. This aspect warrants further investigation in future research.

## F. Robustness of HSPI

Several mitigation strategies can be employed to make the inference of hardware and software platforms significantly more difficult. These strategies focus on disrupting the patterns and subtle variations exploited by HSPI. For logits, introducing random bit flips into the decision-making process can help mask the unique quantization patterns associated with specific hardware architectures. This technique adds a layer of noise that obscures the underlying hardware fingerprints, while incurring minimal overheads associated with sampling random numbers. Similarly, for border inputs, adding random noise to the input data can disrupt the precise calculations that lead to divergent behavior across different hardware and software configurations. This noise injection makes it harder for attackers to identify the specific conditions that trigger variations in model outputs, yet can potentially come with utility degradation.

