# OpenReview forum: "Hardware and Software Platform Inference"
_ICML.cc/2025/Conference — ICML 2025 poster_

### Official Review · Reviewer_ucu1 · 2025-03-08

**Overall Recommendation:** 2

**Summary:**

This manuscript proposed a method called Hardware and Software Platform Inference (HSPI), aiming at identifying the GPU and software stack for machine learning models. The authors introduced 2 methods:
1. HSPI with Border Inputs (HSPI-BI), which is building inputs that are at the decision boundary of the model.
2. HSPI with Logit Distributions (HSPI-LD) uses the distribution of output logits to figure out the hardware environment.
To evaluate ton these methods, the authors employed vision models and language models and used white-box attacks and black-box attacks. The results showed that the proposed method HSPI-LD can figure out GPU types and data types.

**Claims And Evidence:**

No, the proposed method is called "hardware and software platform inference", which I doubt is kind of big regarding the manuscripts, which is mostly focusing on GPU types and data types (like int8, fp16, etc.)
Also, I was wondering about the scalability of the proposed methods. Will they be easily applied to other hardware configurations and software stacks (like different machine learning compilers)?

**Essential References Not Discussed:**

n/a

**Experimental Designs Or Analyses:**

Yes, I feel several experiments' results are not fully and clearly discussed.
Figure 5, different logits showed similar results, what do the authors mean "Figure 5 illustrates the kernel density estimation of various quantization methods where the shapes imply the obvious differences"?
Figure 6, what does it imply?

**Methods And Evaluation Criteria:**

Yes, most of them make sense. I feel several experiments' results are not fully and clearly discussed.
Figure 5, different logits showed similar results, what do the authors mean "Figure 5 illustrates the kernel density estimation of various quantization methods where the shapes imply the obvious differences"?
Figure 6, what does it imply?

**Other Comments Or Suggestions:**

n/a

**Other Strengths And Weaknesses:**

n/a

**Questions For Authors:**

1. Figure 5, different logits showed similar results, what do the authors mean "Figure 5 illustrates the kernel density estimation of various quantization methods where the shapes imply the obvious differences"?
2. Figure 6, what does it imply?
3. I was wondering about the scalability of the proposed methods. Will they be easily applied to other hardware configurations and software stacks (like different machine learning compilers)?

**Relation To Broader Scientific Literature:**

n/a

**Theoretical Claims:**

n/a

---

> ### Author Rebuttal · Authors · 2025-03-29
>
> Thank you for offering valuable questions. We will address them one by one.
>
> > # Claims And Evidence
> > No, the proposed method is called "hardware and software platform inference", which I doubt is kind of big regarding the manuscripts, which is mostly focusing on GPU types and data types (like int8, fp16, etc.) ...
>
> - Our method does go beyond GPUs and data formats. In Tab 2 and Tab 6, we show that **different kernel implementations, runtime libraries, and parallelism strategies**, each produce unique floating-point “fingerprints”. As stated in line 420 page 8 and Tab 10, in practice, we can merge labels and focus on exactly the configurations that we care about—making the approach scalable to new hardware and software stacks.
> - Please refer to our answer to Q3 for details.
>
> > # Q1.
> > Figure 5, different logits showed similar results, what do the authors mean "Figure 5 illustrates the kernel density estimation of various quantization methods where the shapes imply the obvious differences"?
>
> Please zoom in Fig.5. You may notice there are several places where the kernel densities of different data types are clearly not overlapped. For example, the density value at logit value = -1.3 and 0.65, and the whole INT8 curve. We include Fig.5 to help readers intuitively understand how SVMs capture differences in logits distribution.
>
> > # Q2.
> > Figure 6, what does it imply?
>
> Figure 6 visualizes the difference in logit bit distribution between RTXA6000 and A100. We send identical inputs to the same FP16 model checkpoint deployed on RTX6000 (orange) and A100 (blue), and collect outputs. Then we pick the first eight FP32 logits of each output sample (8*32=256 bits per output sample), plot a histogram of the bit counts (the number of ones at that bit) of these 256 bits over all output samples, cut off the minimum bit count of RTX6000 and A100 for each bit, to only show the difference. This implies though we send the same inputs to the same model checkpoints, the bit distribution of outputs from the two GPUs are significantly different even visible to human eyes. We include Fig.6 to help readers intuitively understand why HSPI-LD is possible.
>
> > # Q3.
> > I was wondering about the scalability of the proposed methods. Will they be easily applied to other hardware configurations and software stacks (like different machine learning compilers)?
>
> Yes, we answer this question from three aspects:
> - Hardware configurations
>   - Yes, as we show in Tab R1, HSPI can be easily applies to AMD’s CDNA3 architectures and even **ASIC accelerators** like Amazon’s Inferentia.
>   - In the paper and Tab R2, we already show HSPI works across various GPUs like H100, A100, RTX6000, L40S,  L40, A40, RTX8000, RTX2080Ti, etc.
>   - We mainly run on NVIDIA GPUs in the manuscript because setting up the experiment and running experiments is time consuming.
> - Software stacks
>   - We include results **with and without ML compiler** in the manuscript. Tab 2 runs the LLM inference in the eager mode, while Tab 6 applies a series of optimizations including TorchInductor (torch.compile), cuda graph, etc.
>   - In Tab.R2, we additionally show HSPI can differentiate **different ML compilers (TenorRT-LLM vs TorchInductor)**.
> - Scale up
>   - In Tab.6, we run LLM serving engine with torch.compile, cuda graph, RadixAttention, dynamic batching enabled. HSPI still differentiate different GPUs, even kernel implementations, data types, parallelism strategies.
>   - In Tab.R1, we additionally show HSPI still work after we scale up the setup to **8-GPU system + LLM serving engine + 70B parameter model**.
> We know it is not possible to provide results of all combinations covering all hardwares, but we still tried our best to include as various platform setup as possible.
>
> RTab.1 : HSPI-LD results for Amazon, AMD, NVIDIA (Llama-3.1-70B-it)
> |**Vendor**|**Hardware**|**Software**|**DType**|**Accuracy**|**F1**|
> |-|:-:|:-:|:-:|:-:|:-:|
> |Amazon|Inferentia|NKI|BF16|0.968|0.924|
> ||||FP16|0.988|0.908|
> |AMD|MI300X|ROCm|BF16|1.0|1.0 |
> |||| FP16 |0.906|0.913|
> |NVIDIA|H100|CUDA|BF16|1.0| 1.0 |
> ||||FP16|0.922|0.915|
> ||||Avg|0.964|0.943|
>
> RTab2: HSPI-LD results for large systems (Llama-3.1-70B-it)
> |**Vendor**|**Hardware (Arch)**|**Num devices**|**DP,TP**|**DType**|**Accuracy**|**F1**|
> |--|:---:|:---:|:---:|:---:|:---:|:---:|
> |AMD| MI300X (CDNA3)|2|DP1TP2|BF16|1.0 | 1.0 |
> |||||FP16|0.961|0.957|
> |NVIDIA |H100 (Hooper) | 4 | DP2TP2 | BF16 | 1.0 | 1.0 |
> |||||FP16|0.930| 0.941 |
> ||L40 (Ada)| 8 |DP1TP8 | BF16 | 1.0 | 1.0 |
> |||||FP16 |0.969| 0.958 |
> ||A40 (Ampere)|8| DP1TP8 | BF16 | 1.0 | 1.0 |
> |||||FP16|0.922| 0.925 |
> |||||Avg|0.973| 0.973 |
>
> RTab.3: HSPI-LD results for ML compilers (Llama-3.1-70B-it)
> | **GPU**|**ML Compiler**|**DTypes**|**Accuracy**|**F1**|
> |-|:-:|:-:|:-:|:-:|
> |H100 |TensorRT-LLM | BF16 | 0.984 | 0.927 |
> ||| FP16 | 0.947 | 0.944 |
> || TorchInductor | BF16 | 1.0 | 1.0 |
> ||| FP16 | 0.930 | 0.941 |
> ||| Avg | 0.965 | 0.953 |
>
> Please let us know if you have further questions :)

---

### Official Review · Reviewer_A46R · 2025-03-12

**Overall Recommendation:** 3

**Summary:**

The paper introduces Hardware and Software Platform Inference (HSPI), a novel method for identifying the underlying GPU architecture and software stack of machine learning models based on their input-output behavior. HSPI uses computational differences across various GPUs and software environments to detect the specific device utilized for inference. The authors present two techniques—HSPI with Border Inputs (HSPI-BI) and HSPI with Logits Distributions (HSPI-LD) and both demonstrate high accuracy rates in white-box and black-box settings while discussing the limitations of the methods.

**Claims And Evidence:**

The experiments support the submission, showing high accuracy for different algorithms, kernels, datatypes, and sharding techniques.

**Essential References Not Discussed:**

No essential work that is unreferenced is noticed.

**Experimental Designs Or Analyses:**

The experimental designs appear sound, with plenty . One criticism would be the model size limitation presented in the experiments (See Other Strengths And Weaknesses).

**Methods And Evaluation Criteria:**

The proposed method and evaluation make sense for the problem, with various benchmark datasets for vision and language tasks strengthening the evaluation criteria. The datasets used are also standard datasets in the correspondent areas.

**Other Comments Or Suggestions:**

No other suggestions.

**Other Strengths And Weaknesses:**

Strength:
1. Novel approach to identifying hardware and software platforms.
2. Clear organization and well-written.

Weakness:
1. The paper states that limited GPU memory would constrain the ability to scale to larger language models; even with HSPI-LD the experiments were conducted on rather small LLMs. Yet, the motivation is more about checking whether a very large model is indeed deployed instead of using a smaller, distilled model. (A possible solution could be combining HSPI with distributed methods.
2. The paper should discuss the computation efficiency of the method.

**Questions For Authors:**

1. Can other common metrics, such as the wall clock time for inference, be useful for identifying the hardware and software stack?
2. Is it possible that different combinations of hardware and software lead to the same prediction?

**Relation To Broader Scientific Literature:**

The paper presents a new problem of verifying the hardware and software of ML models. Prior work on hardware detection methods have lower accuracy rates.

**Theoretical Claims:**

No proofs were presented in the paper.

---

> ### Author Rebuttal · Authors · 2025-03-29
>
> Thank you for valuable suggestions and questions. We would like to address them one by one.
>
> > # W1. Large models and systems
> > The paper states that limited GPU memory would constrain the ability to scale to larger language models; even with HSPI-LD the experiments were conducted on rather small LLMs. Yet, the motivation is more about checking whether a very large model is indeed deployed instead of using a smaller, distilled model. (A possible solution could be combining HSPI with distributed methods.
>
> Yes, large scale experiments are important for evaluating HSPI.
> - In Tab 6 on page 8, we distributed Qwen-2.5-14B across four-GPU systems using the popular LLM serving engine SGLang and enabled a series optimizations including custom cuda kernels, RadixAttention, torch.compile, cuda graph, smart request batching, etc. HSPI works well and even **differentiated different parallelism strategies like DP2TP2 vs DP4TP1**. We assume HSPI-LD will work on larger distributed systems because the communication between devices does not introduce noise to logits distribution.
> - To verify this, we ran a new group of experiments where **we distributed Llama-3.1-70B across a larger multiple high-end GPU systems using SGLang**. As shown in the following table, HSPI-LD still achieves an average accuracy over 97%.
>
> RTab2: HSPI-LD results for large systems (Llama-3.1-70B-it)
> | Vendor | Hardware (Arch) | Num devices | DP,TP | DType | Accuracy | F1 |
> |---|:---:|:---:|:---:|:---:|:---:|:---:|
> | AMD | MI300X (CDNA3) | 2 | DP1TP2 | BF16 | 1.000 | 1.000 |
> |  |  |  |  | FP16 | 0.961 | 0.957 |
> | NVIDIA | H100 (Hooper) | 4 | DP2TP2 | BF16 | 1.000 | 1.000 |
> |  |  |  |  | FP16 | 0.930 | 0.941 |
> |  | L40 (Ada) | 8 | DP1TP8 | BF16 | 1.000 | 1.000 |
> |  |  |  |  | FP16 | 0.969 | 0.958 |
> |  | A40 (Ampere) | 8 | DP1TP8 | BF16 | 1.000 | 1.000 |
> |  |  |  |  | FP16 | 0.922 | 0.925 |
> |  |  |  |  | Avg | 0.973 | 0.973 |
>
> > # W2. Computation efficiency
> > The paper should discuss the computation efficiency of the method.
>
> Yes, we will add the following discussion into the revised version.
>
> In our experiments, HSPI-LD is the most costly method and the cost of HSPI-LD mainly consists of three parts:
> - Collecting training samples: Collecting training samples is the most computationally expensive and slow part. To collect samples, we deploy models on a specific HW-SW setup, run model inference and dump output logits. The cost can be estimated as `num of platform options * num of samples * sequence length * cost of a forward pass`. We spend over 600 GPU hours on this.
> - Training HSPI classifier: As stated in Sec 4.3, we train $N(N-1)/2$ binary SVM classifiers to perform HSPI's $N$-class classification. Thus cost is proportional to `N(N-1)/2 * num of samples`. We use sklearn to train SVMs on CPUs, which is fast as the training iteration is around 1000. Usually the training of all binary classifiers takes less than 20 minutes.
> - Evaluating HSPI classifier: The evaluation feed output logits to all binary classifiers, thus the cost is also proportional to `N(N-1)/2 * num_samples`. The prediction is also on CPU and the cost is ignorable compared to collecting training samples. Usually the prediction takes less than 5 minutes.
>
> For HSPI-BI, the cost of training a batch of border inputs is twice as normal model training because we need to run the forward-backward pass for a pair of platforms. Since we mainly run this methods for CNN models, the GPU hours were much smaller than HSPI-LD.
>
> > # Q1.
> > Can other common metrics, such as the wall clock time for inference, be useful for identifying the hardware and software stack?
>
> Yes it is possible. Wall clock time can be used for creating hardware fingerprints, but we assume wallclock time is less reliable in this context. Deep learning serving engines like TensorRT, vLLM and SGLang may use some dynamic batching strategies which means the wall clock time changes with the request arrival rates. For example, the daytime wall clock time fingerprint is different from the one at night because the server is busier during the day.
>
> > # Q2.
> > Is it possible that different combinations of hardware and software lead to the same prediction?
>
> Yes, it is possible. This is one limitation we discussed in Sec 6.3 and A.4. During experiments we found when the software stack is identical, HSPI cannot differentiate RTX8000 and RTX2080Ti. This is because these two GPUs have both NVIDIA compute capability = 7.5 (Turing architecture), and a very similar number of Tensor Cores around 550 (they fall into the same EQC when the software stack is the same). For more details please refer to Sec 6.3 and A.4.
>
> Please let us know if you have further questions :)

---

> > ### Comment · Reviewer_A46R · 2025-04-07
> >
> > Thank you for your detailed response. I think the work is solid, and it's novel in identifying the hardware and software based on the input-output. I will keep my original score, partially due to my unfamiliarity with this line of work.

---

### Official Review · Reviewer_WXiw · 2025-03-13

**Overall Recommendation:** 3

**Summary:**

The paper presents an interesting idea where a client can infer the hardware and software platform that was used for model inference based on its input and output behavior. It banks on the observation that there are inherent differences between different GPUs and software stack. The idea has potential to allow clients to verify the actual hardware that was used for inference against malicious service providers.

**Claims And Evidence:**

Besides some questions about the robustness of the approach, I think the paper seems convincing.

**Essential References Not Discussed:**

The paper does not seem to shed enough light on the line of work where side channels are exploited to infer the model and the hardware.
1. Hua, Weizhe, Zhiru Zhang, and G. Edward Suh. "Reverse engineering convolutional neural networks through side-channel information leaks." Proceedings of the 55th Annual Design Automation Conference. 2018.
2. Gongye, Cheng, et al. "Side-channel-assisted reverse-engineering of encrypted DNN hardware accelerator IP and attack surface exploration." 2024 IEEE Symposium on Security and Privacy (SP). IEEE, 2024.

Especially, given that the 2nd paper mentioned above is suggesting that side-channels can be exploited to reverse-engineer the hardware IP, it seems like there are some similarities. It would benefit the reader if the authors can give a description about how the approach and applications may differ.

**Experimental Designs Or Analyses:**

Seems like the approach seems valid. However, there are some questions regarding its robustness and scalability.

**Methods And Evaluation Criteria:**

Besides the fact that the evaluated hardware and software platforms are not diverse enough, it seems the paper tried to evaluate across various inference scenarios.

**Other Comments Or Suggestions:**

The paper presents a simple yet nice idea to infer hardware and software platforms during inference. I think the paper provides a large set of experiments for various scenarios. I would like to stay positive about the paper. However, it would be great to see how the proposed HSPI approach would scale for larger set of hardware and software platforms. Also, would love to understand the potential ways to make this more robust.

**Other Strengths And Weaknesses:**

.

**Questions For Authors:**

1. I am very curious how a client would differentiate between "deviations" from hardware and "deviations" from software. In practice different generations of hardware that have different numerical behavior may return same output due to software optimizations. It seems that 4.1 mentions that these variations are included in H. Can you provide more details in this respect?
2. As the paper states in A.5, the model inference could have introduced some noise/perturbations (consider the fact that different inference may return different output despite the same model). In that case, how would the client filter out that noise to perform HSPI in a robust manner
3. It would be interesting to investigate what features were used to distinguish different precision, kernel, hardware, ... Interpretation of the HSPI model would provide some interesting insights.
4. It would be more interesting to see experimentatal results for more variegated class of accelerators (not just GPUs) considering their prevalence (TPUs, Inferentia, MTIA, Maia100).
5. For the black-box access only scenario in 4.1, client might not have the full picture considering that the service providers may have appended prompt for LLM inference. How is the HSPI impacted and what may be mitigations to make HSPI more robust?
6. Service providers may limit the inference requests while the client tries to infer the hardware, how would this be distinguished from the DoS attack.
7. How does the paper scale tolarger set of hardware and software platforms. Seems like there is a potential for signficant drop in accuracy of the HSPI.

**Relation To Broader Scientific Literature:**

The paper presents a simple yet insightful extension to the line of work where we have used input/output data to infer the model and the hardware architecture, and various defense mechanisms against these side channel attacks and IP infringement.

**Theoretical Claims:**

Seems the given equations make sense. There are no theoretical proofs for it.

---

> ### Author Rebuttal · Authors · 2025-03-29
>
> Thank you offering valuable suggestions and questions. We will address them one by one. Due to word limits, we uploaded **three new result tables** here: https://imgur.com/a/VbONCoU
>
> > # E1. Essential References Not Discussed
>
> Our work is different from these two papers in terms of goals, threat models (methods), and generalizability.
>
> Goals
> - Ref 1 extracts information about CNN models, e.g, whether the model is AlexNet or SqueezeNet, from an FPGA-based CNN accelerator, by analyzing memory access patterns.
> - Ref 2 predicts information about the encrypted IPs on an FPGA-based DNN accelerator. They also recover characteristics about model architectures and extract model params.
> - Our work aims to predict information about the SW-HW stack used for serving large deep learning models, including compiler, data types, GPU arch, parallelism strategy, etc. **We do not aim to predict which model** is being served.
>
> Threat models
> - Ref 1 and 2 assume side channel information about the HW like off-chip memory access patterns, electromagnetic traces, schematics is accessible.
> - We assume the SW-HW platform serves DNNs in the cloud and **we only have access to the request and responses via the service provider’s API**.
>
> Generalizability
> - Ref 1 and 2 tests their methods on a specific FPGA accelerator.
> - We tried our best to test the generalizability of HSPI, across model families, data types, GPU archs, parallelism strategies, etc.
>
> We will include these two papers in our related work section and discuss the differences there.
> > # Q1
> > I am very curious how a client would differentiate between "deviations" from HW and "deviations" from SW ...
>
> One can differentiate deviations caused by HW and SW because they will be in different EQCs and we can enumerate all of them by going through all HW and SW combinations. However, in Sec 6.3 and A.4  we do get the same output from RTX8000 and RTX2080Ti (they both have NVIDIA compute capability = 7.5 and very similar number of Tensor cores ~550) when using the same SW stack, thus these two combinations are left in the same EQC. Please refer to 6.3 and A.4 for more details.
> > # Q2
> > As the paper states in A.5, the model inference could have introduced some noise/perturbations ...
>
> One would need to increase the number of queries sent by HSPI to capture statistics in different HW and SW settings (the cost increases, too). On the other hand, the introduced noise reduces the model performance, e.g, LLM generates low-quality texts.
> > # Q3
> > It would be interesting to investigate what features were used to distinguish different precision, kernel, HW, ...
>
> Our method is based on the EQC theory explained in Sec 2. Limited by rebuttal word counts, we would recommend [this paper](https://openreview.net/forum?id=6zyFgr1b8Q), in which they present a more detailed analysis identifying how architectural choices impact computational stability and precision deviations.
> > # Q4
> > It would be more interesting to see experimentatal results for more variegated class of accelerators ...
>
> - Yes, we are also curious about this. We believe our observation on NVIDIA GPUs still holds on other accelerators because there may be more differences captured by HSPI classifiers, such as different accumulator length in compute units. Unfortunately, limited by resources, we are not able to test HSPI on all these platforms.
> - We run new experiments including Amazon Inferentia and AMD Instinct MI300X (**RTab.1 in the link**). HSPI still successfully differentiates them (avg.acc=96.4%).
>
> > # Q5
> > For the black-box access only scenario in 4.1, client might not have the full picture considering that the service providers may have appended prompt ...
>
> - During experiments, we already prepend chat prompts before our request. For example, each of our Qwen-2.5 input concatenates “You are Qwen, created by Alibaba Cloud. You are a helpful assistant” and our query “Please generate <num_random_words> random words …” . We find using these simple chat prompts actually helps to generate more random outputs.
> - For complex scenarios, HSPI can be combined with prompting tricks like jailbreaking to be more robust.
>
> > # Q6
> > Service providers may limit the inference requests ... DoS attack.
>
> Commercial inference services are designed to support high‑throughput access. Moreover, HSPI does not need to collect all the responses in a very short period, so request rates of HSPI are lower than the limits set by service providers, lower than DoS attacks by order of magnitudes. For example, in Tab 6 experiments, we sent 6 requests/sec, and our method collects 256 inference queries in around 40 minutes, which is comparable to normal application workloads. We can also distribute queries across multiple accounts in case of strict rate limit.
> > # Q7
> > How does the paper scale to larger set of HW and SW platforms...
>
> We ran new large scale experiments (**RTab2** in the link) and HSPI still works well. Please refer to our answer to Reviwer A46R.

---

### Official Review · Reviewer_4ddm · 2025-03-14

**Overall Recommendation:** 3

**Summary:**

This paper introduced Hardware and Software Platform Inference (HSPI), which is a method for identifying the hardware and software stack based on the input-output behavior of machine learning models. The proposed method leverages the inherent differences of various GPU architectures and compilers to distinguish between different GPU types and software stacks. Experiment show that in a white-box setting the method can distinguish between different GPUs with between 83.9% and 100% accuracy; in a black-box setting the method can achieve results that are up to 3 times higher than random guess accuracy.

## update after rebuttal
Questions are answered, I remain positive about this work and would like to keep my score.

**Claims And Evidence:**

1. HSPI is possible because of the different characteristics of hardware and software configuration -- this is supported by the analysis in section 2 and examples showed in Figure 2, Figure 3 and Figure 5.
2. HSPI can effectively distinguish different GPUs with high accuracy in both white-box and black-box setting -- this is supported by results demonstrated in Table 1-4.

**Essential References Not Discussed:**

N/A.

**Experimental Designs Or Analyses:**

1. The factors made HSPI possible discussed in section 2 makes a lot sense, and demonstrate good insights into low level implementation of machine learning models.
2. Experiments including white-box and black-box, authors proposed to compare the identified quantization method, GPU type, and kernels used to ground truth, and compare accuracy / F1 score. The scores demonstrated its capability to identify underlying hardware and software choice, with high accuracy.

**Methods And Evaluation Criteria:**

Yes the authors proposed to use different combinations of models, GPUs, kernels, and quantization method to conduct white-box and black-box experiments against HSPI, and collect accuracy / F1 score to show its effectiveness. Details was demonstrated in Table 1-6.

**Other Comments Or Suggestions:**

N/A.

**Other Strengths And Weaknesses:**

This paper provided a very new perspective to look at current model hosting hardware and software stack, especially considering the quantization methods and white-box vs. black-box scenarios.

**Questions For Authors:**

1. Given the current method for hardware and software platform inference, how to avoid the stack to be exposed to users?

**Relation To Broader Scientific Literature:**

This paper is based on the ideas from Schlögl et al. (2023) where it targeted CPU identification based on the same equivalence classes. And further, the authors considered Computational Deviations, and focused on GPUs with both white-box and black-box settings, proposed an effective solution to identify the underlying hardware / software stack with high accuracy.

**Theoretical Claims:**

N/A.

---

> ### Author Rebuttal · Authors · 2025-03-29
>
> Thank  you for offering valuable suggestions and questions. We would like to address them one by one.
>
> > S1. The authors shared their code to training the classifier and run inference. We don't have the logit dataset to run but it looks legit.
>
> - We know collecting the logits from various combinations of `[model checkpoint, hardware stack, software stack]` are resource consuming, so we will open-source the logits datasets once the paper is accepted.
> - The supplementary materials also include the codes for producing the logit dataset. For example, in line 227 of `llm_logits_svm.py`, the function `create_logits_dataset` collects logits for training the classifier.
>
> > Questions For Authors:
> > Q1. Given the current method for hardware and software platform inference, how to avoid the stack to be exposed to users?
>
> There are several possible ways but may have side effects on user experience like lower generation quality and longer latency.
> - In Appendix 5, we suggested random bit flips and adding random noise as potential ways of mitigation. For further explanation, please refer to Appendix 5 on page 13.
> - Furthermore, adding other forms of randomizations such as reordering arithmetics randomly or switching between multiple compilation kernels would make our method more expensive, since then a statistical average would need to be calculated on a lot more samples to see a difference in distributions between different hardware and software platforms.
> - Lastly, limiting model logit access, also would make it harder for HSPI. However, the logits is an important part of API enabling users to explore smart sampling strategies like test-time scaling.
>
> We will add this discussion to the revised manuscript.
>
> Please let us know if you have further questions :)

---

> > ### Comment · Reviewer_4ddm · 2025-04-08
> >
> > Thanks for the response, good to know the several ways to avoid exposure of stack details, I think those would be great to be added to appendix. And I remain positive about this work.

---

### Decision · Program_Chairs · 2025-05-01

**Decision:**

Accept (poster)

**Comment:**

**Meta-Review of “Hardware and Software Platform Inference”**

This submission tackles the problem of verifying the actual hardware and software stack used to serve a machine learning model by analyzing its input–output behavior. The core idea is that different GPU architectures, data types (precision/quantization), kernel implementations, and runtime optimizations lead to subtle but detectable numerical patterns in the model’s output. The authors introduce two methods:

1. **HSPI-BI (Border Inputs)**: Creates specialized inputs that lie at the decision boundary between two (or more) model configurations, thereby eliciting different predictions depending on the serving hardware/software.
2. **HSPI-LD (Logits Distributions)**: Trains a classifier (e.g., an SVM) on the distribution of model logits, splitting and analyzing the floating-point bits or entire logit distributions to distinguish different hardware/software setups.

They evaluate both methods using vision models (e.g., ResNet, MobileNet) and language models (e.g., LLaMA variants, QWen, Phi). The experiments span “white-box” scenarios (where the model weights and architecture are known and can be directly instrumented) and “black-box” scenarios (where only the model outputs—particularly logits—are accessible). Their primary findings:

- **White-box**: HSPI-BI can reliably separate GPUs of distinct compute capabilities (e.g., A100 vs. A6000) or different quantization schemes (FP32, BF16, INT8, etc.) and often achieves near-perfect classification accuracy. HSPI-LD can also reach high accuracy (up to 100%) for distinguishing hardware platforms or software configurations, and it scales better than HSPI-BI for large language models.
- **Black-box**: While performance is lower, the authors report that the approaches (particularly HSPI-LD) still achieve up to three times random-guess accuracy for certain configurations—demonstrating that even with minimal visibility into the model, hardware inference remains possible.

---

## Strengths

1. **Hardware and Software Platform Inference (HSPI)**:
   - Proposes important problem of verifying that an ML service is actually running on the promised hardware/software configuration. This is increasingly relevant in commercial and security contexts.
   - The authors discuss how their method can aid in verifying hardware compliance and security, aligning well with the ongoing push for more transparency and trust in AI supply chains.

2. **Algorithms**: Two complementary approaches (HSPI-BI and HSPI-LD) that exploit numerical discrepancies in floating-point arithmetic and quantization.

3. **Extensive Experiments**:
  - Systematic evaluations using computer vision and large language models, covering multiple GPUs (A100, L40, A6000, etc.), quantization levels, parallelization strategies, and ML compilers. The experimental evidence demonstrates that HSPI can succeed across diverse settings.
  - In white-box mode, near-perfect accuracy is often achieved (≥ 99% in many cases). Even in black-box mode, they show that accuracy can substantially exceed random guessing.

---

## Weaknesses

1. **Limited Black-Box Robustness**
   - While the black-box results are promising relative to random guess, they remain significantly lower than the near-perfect rates in white-box settings. The method can require collecting many queries, and performance might degrade if providers introduce noise or limit logit access.

2. **Identical or Very Similar GPU Architectures**
   - If two platforms reside effectively in the same “equivalence class” (e.g., nearly identical compute capabilities), HSPI may fail to differentiate them. The paper acknowledges this (e.g., RTX 8000 vs. RTX 2080 Ti) as a practical limitation.

3. **Scalability**
   - Generating border inputs (HSPI-BI) for larger LLMs can become GPU-memory-intensive due to the need to store full training activations and run backward passes. HSPI-LD is more scalable but requires collecting and classifying logits, which can still impose overhead or costs in pay-per-token settings.

4. **Comparison with Broader Side-Channel Literature**
   - Although the paper references prior hardware fingerprinting and side-channel work, reviewers noted that more in-depth contrast with existing CPU/GPU side-channel attacks (or IP reverse-engineering approaches) could strengthen the paper.

---

## Recommendation

Based on the paper's contributions I recommend accepting this paper to ICML.